# Nucleolar sequestration of cannabinoid type-2 receptors in triple-negative breast cancer cells

Linley P. Prado-Celis[1], Rodrigo Zamora-Cárdenas[1], Javier Alamilla[1,2], Enrique A. Sánchez-Pastor [1], Tania Ferrer[1], Eloy G. Moreno-Galindo[1]*, Ricardo A. Navarro-Polanco[1]*

1 Centro Universitario de Investigaciones Biomédicas "CUIB", Universidad de Colima, Colima, Colima, Mexico, 2 Consejo Nacional de Humanidades, Ciencia y Tecnología (CONAHCYT), Universidad de Colima, Colima, Mexico

* magdal@ucol.mx (RANP); eloy@ucol.mx (EGMG)

## Abstract

Multiple investigations have shown that the different types of cannabinoids, phyto-cannabinoids, synthetic cannabinoids, and endocannabinoids, possess antiprolifer-ative and anticancer properties. The cannabinoid type-2 receptor (CB2R) has been proposed as a central player in tumor progression and has been correlated with the aggressiveness of breast cancer. Using immunocytochemistry and confocal micros-copy, in the present work, we studied the expression level and subcellular localization of CB2R in two human triple-negative breast cancer (TNBC) cell lines, corresponding to early (stage I, HCC-1395) and metastatic (MDA-MB-231) stages, and they were compared with a non-tumoral mammary epithelial cell line (MCF-10A). We found that although CB2R was detected at the plasma membrane, it was mainly localized intra-cellularly, with ~40-fold higher expression in both TNBC cell lines than in MCF-10A ($P < 0.0001$). Notably, double staining with DAPI or with the nucleoli-specific fluores-cent marker (3xnls-mTurquoise2) showed that most of the CB2R overexpressed in the nucleoli of cancer cells. This finding is supported by the fact that CB2R expres-sion was markedly lower in mitotic cells compared to interphase cells ($P < 0.0001$). Interestingly, exposure of cancer cells to the specific agonist HU-308 reversed the nucleolar sequestration of CB2R while increasing the presence of the receptor in the nucleoplasm and cytoplasm ($P < 0.0001$). In addition, we found that this agonist reduced both the cell migration ($P < 0.05–0.0001$) and proliferation ($P < 0.001$) of TNBC cells. It remains to determine the function and signaling ability of CB2R in the nucleolus. Although our study only includes cell lines (tumoral and non-tumoral), we consider that this feature of nucleolar sequestration of CB2R could be a potential diagnostic marker for TNBC from the early stage.

**Data availability statement:** All relevant data are within the manuscript and its Supporting Information files.

**Funding:** This work was funded by Consejo Nacional de Humanidades, Ciencia y Tecnología (CONAHCYT), Mexico to: R.A.N-P., E.G.M-G., and J.A. (Grant No. FC2015-1-121); R.A.N-P. (Grant No. 2023-G-1340); and Infrastructure Grant No. 321696. L.P.P-C. and R.Z-C. were supported by Doctoral Fellowships (No. CVU 933434 and 618681, respectively) from CONACYT, Mexico. The funders had no role in study design, data collection and analysis, decision to publish, or preparation of the manuscript.

**Competing interests:** The authors have declared that no competing interests exist.

## Introduction

Breast cancer represents the highest incidence rate of cancer in women worldwide. In recent years, there has been an increase in the rate of more aggressive breast cancer subtypes, particularly in young women [1,2]. Among these is triple-negative breast carcinoma (TNBC), so named due to its lack of estrogen, progesterone, and epidermal growth factor type-2 receptors. The absence of these receptors in TNBC reduces pharmacological targets for medical treatment, thus contributing to the strong aggressiveness and poor prognosis that characterize this type of cancer [3].

It has been shown that phytocannabinoids, endocannabinoids, and synthetic cannabinoids possess antitumor properties [4–7]. Most of these effects are mediated by the endocannabinoid system, which is composed of: receptors (mainly cannabinoid type 1 and 2 receptors (CB1R and CB2R, respectively)), endocannabinoids (lipid signaling molecules, such as anandamide and 2-Arachidonoylglycerol), and the enzymes that synthesize and degrade endocannabinoids [8]. CB2R is mainly expressed in peripheral tissues and is up-regulated in cannabinoid-mediated anti-inflammatory effects and immune response regulation [4]. Furthermore, several studies have shown that CB2R plays an important role in tumor progression, and the expression level of this receptor in breast cancer correlates with the aggressiveness of tumors [4,6]. Indeed, CB2R expression is increased in TNBC [3,5,9–11]. Thus, efforts to develop new therapies using CB2R as a pharmacological target are augmenting.

Recently, intracellular expression of CB2R was reported in both pyramidal neurons and epithelial cells of human osteosarcoma, where it modulates $Ca^{2+}$ signaling by coupling to Gq proteins, suggesting that intracellular CB2R has a different physiological role from that of the cell membrane, where it is mainly associated with Gi [12,13]. However, it is unknown whether the overexpression of CB2R in breast cancer cells entails the presence of this receptor at the intracellular level in addition to the plasma membrane. Therefore, here we set out to identify whether TNBC cells express internalized CB2R and its possible association with intracellular structures. Using confocal microscopy and immunocytochemical (ICC) approaches, we found that although CB2R is overexpressed in TNBC cells, there is a massive accumulation of this receptor within the nucleoli. The CB2R nucleolar sequestration was reversed by the specific agonist HU-308, which also negatively affected the cell migration and proliferation of TNBC cells. The extraordinary nucleolar sequestration of CB2R in TNBC cells is a novel finding with potential applications in the diagnosis of this pathology.

## Materials and methods

### Cell culture and transfection

For the comparative study of the expression and subcellular localization of CB2R, we used a non-cancerous human breast epithelial cell line (MCF-10A), an early-stage (stage I; HCC-1395), and a highly metastatic (MDA-MB-231) TNBC cell lines, as good representative models of TNBC progression. The cell lines were purchased from American Type Culture Collection (ATCC, Manassas, VA, USA). HCC-1395 and MDA-MB-231 cells were cultured in RPMI 1640 medium (Sigma-Aldrich, St. Louis,

MO, USA), while MCF-10A cells were cultured in DMEM/F12 medium (Gibco; Thermo Fisher Scientific Inc., Waltham, MA, USA). Both media were supplemented with 10% fetal bovine serum (Gibco) and 1% antibiotic-antimycotic solution (Sigma-Aldrich;100 units penicillin, 100 µg streptomycin, and 0.25 µg amphotericin B per milliliter). Additionally, insulin (10 µg/mL), hydrocortisone (0.5 µg/mL), and human epidermal growth factor (20 ng/mL) (the three from Sigma-Aldrich) were added to the DMEM/F12 medium. Cells were grown in 60-mm tissue culture dishes (Corning Inc., Corning, NY, USA) and maintained at 37 °C in an incubator with a humidified atmosphere and air/5% $CO_2$. Cells were used between 3–29 passages.

For transfection of the plasmid to express the blue fluorescent protein (3xnls-mTurquoise2), HCC-1395 and MDA-MB-231 cells were seeded on 18 × 18 mm coverslips and transfected with this plasmid (3 µg), which was a gift from Dorus Gadella [14] (Addgene plasmid #98817; http://n2t.net/addgene:98817; RRID: Addgene_98817), using Lipofect-amine 2000 (Invitrogen; Thermo Fisher Scientific Inc., Waltham, MA, USA) 24 h before ICC assays. Likewise, for treat-ment with the CB2R-specific agonist, HU-308 was directly added to the pre-seeded cells. (DMEM/F12 or RPMI 1640) for a 24-h incubation period before the ICC protocols.

## Immunocytochemistry

Cells were fixed in 4% paraformaldehyde diluted in PBS (pH 7.4) for 10 min at room temperature (22–24 °C). Then, cells were permeabilized with 0.1% Triton X-100 in PBS for 30 min at room temperature, blocked with 10% donkey serum in PBS for 1 h at room temperature, and incubated overnight at 4 °C with rabbit polyclonal Anti-Human CB2R (1:500, Alomone Labs, Jerusalem, Israel; ACR-003) diluted in PBS. Afterwards, cells were washed in cold (4 °C) PBS (3 times for 5 min each) and incubated for 2 h at room temperature with the secondary antibody (1:1000, donkey Anti-rabbit IgG; Abcam, Cambridge, MA, USA; ab150075) conjugated with Alexa Fluor 647 (Abcam). After three washes in cold PBS, samples were mounted using ProLong™ Diamond Antifade Mountant with DAPI (Thermo Fisher Scientific). Antibody characterization

The anti-human CB2R antibody is a polyclonal antibody raised in rabbit, (Alomone Cat# ACR-003). The immunizing antigen peptide sequence is (C)NGSKDGLDSNPMKD, corresponding to the amino acid residues 11–24 of the human CB2R, located in the extracellular N-terminus, as provided by the vendor company. Immunoblots analysis by the supplier revealed three bands corresponding to 100, 37, and 20 kDa; similar to previous findings [15]. Immunoblots performed with the blocking peptide showed no immunostaining in experiments using lysates from HL-60 acute promyelocytic leukemia cells and human MCF-7 breast adenocarcinoma cell line (product datasheet). This result was confirmed in the current study, where the specificity of the CB2R antibody was tested by incubating it for 2 h at 37 °C in the presence of its specific blocking peptide (at 1:3 ratio; Alomone Cat# BLP-CR003) (S1 Fig).

## Image acquisition and analysis

Images were acquired under a laser scanning confocal microscope (Axio Observer LSM700; Carl Zeiss, Oberkochen, Germany), using a 40 × or 63 × oil objective (NA 1.4), with no optics zoom. Every image consisted of approximately 20 consecutive optical slices (Z-axis). The acquisition parameters (photomultiplier gain and laser intensity) were established according to the cell line that exhibited the highest brightness (MDA-MB-231) without saturation, and thus these parame-ters were used for the image acquisition of the other cell lines studied. Cancerous and non-cancerous cells were stained and imaged simultaneously for consistency between runs to minimize any artifact differences. The images were obtained in the lowest possible time, minimizing the chances of artifact differences due to fluorescence rundown. Both the acqui-sition and image analysis of the cell lines studied were done on a random basis by blinded observers/analyzers. Images were analyzed with ImageJ software (NIH, Bethesda, MD, USA) and split into two channels, while the integrated density evaluation was used for fluorescence measurement, and expressed as relative fluorescence units (RFU).

## Wound-healing assay

HCC-1395 and MDA-MB-231 cell lines (~70–80% confluence) were cultured in 12-well plates. Using a sterile 200 µL-micropipette tip, a linear scratch/wound was created in the cell monolayer in each well. After that, images were taken every 3 h over a time-lapse between 0 and 36 h using a Nikon Diaphot 300 inverted microscope (Nikon Instruments Inc., Melville, NY, USA) at 4 × magnification. Experiments were performed in triplicate in both control (vehicle (< 0.001% ethanol)) and HU-308 agonist (10 µM) groups. Using ImageJ software (NIH), the relative wound closure was calculated as: $A_t/A_0$, where $A_0$ is the initial area of the wound, while $A_t$ is the area measured at the end of each 3-h interval.

## Cell proliferation assay

Cell proliferation was assessed using the Click-iT EdU Alexa Fluor 647 Kit (Invitrogen) by triplicate. Briefly, MCF-10A and TNBC cell lines were cultured on poly-D-lysine-coated 18-mm coverslips. Cells were incubated with the vehicle (< 0.001% ethanol) or 10 µM HU-308 for 24 h. After incubation, 5-ethynyl-2'-deoxyuridine (EdU) was added at a final concentration of 10 µM for 2 h. The cells were then fixed with 4% paraformaldehyde, permeabilized with 0.5% Triton X-100, and revealed with the reaction buffer. Finally, the samples were mounted using ProLong™ Diamond Antifade Mountant with DAPI (Thermo Fisher Scientific). Images were acquired with the Axio Observer LSM-700 confocal microscope (Carl Zeiss) with a 40 × objective. The proliferation percentage was calculated as the number of EdU-positive nuclei divided by the total number of nuclei in each image, and multiplied by 100.

## Statistical analysis

Data are expressed as the median and interquartile range (IQR) and as the mean ± SEM, as appropriate. The distribution of data was checked with the Shapiro–Wilk and Kolmogorov-Smirnov tests. The significance of differences was assessed by the unpaired $t$-test and the non-parametric Mann-Whitney U-test and Kruskal-Wallis test followed by the Dunn's post hoc test, as required. Statistical analysis was performed using Prism 8 software (GraphPad Software, La Jolla, CA, USA). Differences were considered to be significant at $P < 0.05$.

## Results

### TNBC cells overexpress CB2R intracellularly, mainly in the nucleus

To assess the overall expression and distribution of CB2R both in the plasma membrane and in the intracellular compartment of tumoral and non-tumoral mammary cell lines, we first assayed the presence of CB2R in two TNBC cell lines, one of the initial stage (HCC-1395) and another considered highly metastatic (MDA-MB-231), as well as in a non-cancerous mammary epithelial cell line (MCF-10A). Compared to the latter, cell-wide cannabinoid receptor expression was markedly higher in cancer cells by approximately 40-fold: 1.8 RFU [IQR: 4.1–1.4 RFU] in MCF-10A *versus* 77.2 RFU [IQR: 97.0–62.8 RFU] in HCC-1395 and 69.1 RFU [IQR: 136.9–55.1 RFU] in MDA-MB-231 cells (p < 0.0001; Kruskal-Wallis test) (Fig 1A and B). Of note, although CB2R was perceived at the plasma membrane, this protein was predominantly detected at the intracellular level, principally in the nuclear region (Fig 1A and B), where the staining appears to be especially intense in the nucleoli. By contrast, in non-permeabilized cells, the CB2R signal was mainly restricted to the plasma membrane and was virtually absent inside the cell (especially in the nucleus) (Fig 1C).

### TNBC cells accumulate CB2R in the nucleoli

To examine the apparent localized presence of CB2R in the nucleolus of TNBC cells, we analyzed the emission intensity profile in a cross-section of the nucleus, both with the DAPI and CB2R channels (Fig 2A). As shown in the figure, the DAPI staining identifies the nucleus (grey) while the nucleolus appears as a black hole, and thereby in this region the DAPI signal intensity clearly decreased to the basal level (Fig 2A, left). On the contrary, staining with the anti-CB2R antibodies

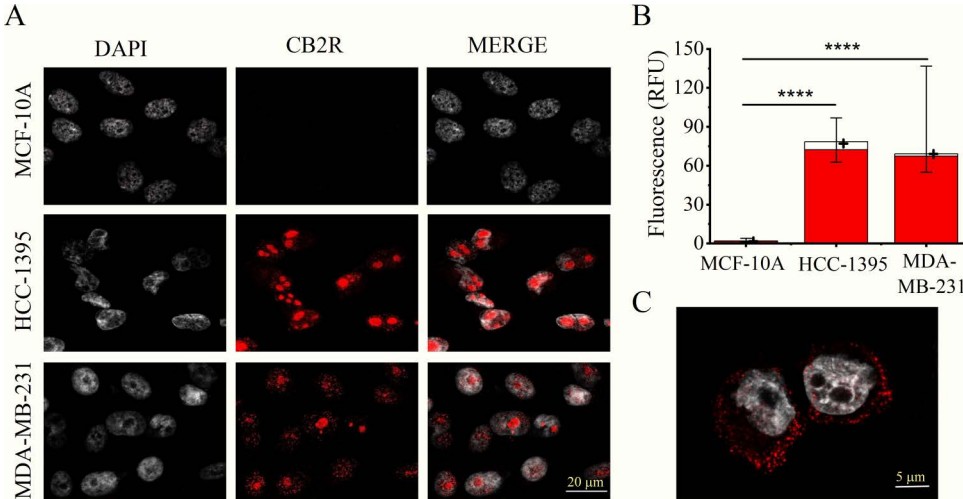

**Fig 1. TNBC cell lines overexpress CB2R. A**, Representative ICC confocal images of non-tumoral (MCF-10A) and TNBC (HCC-1395 and MDA-MB-231) cell lines. Images of the left column show the fluorescence of the DAPI signal (gray) corresponding to the cell nucleus. The middle column depicts the signal obtained with a specific antibody against CB2R (red, Alexa 647); and the right column shows the merged images. **B**, Fluorescence intensity (arbitrary units (RFU)) of CB2R measured as the total cell fluorescence (empty bars) and only in the cell nucleus (red bars). The empty bars denote the median values, and the error lines display the Q3 and Q1 quartiles. The small plus sign inside the bars indicates the mean value. **C**, Representative CB2R immunostaining in non-permabilized HCC-1395 cells: note that the CB2R signal is mainly restricted to the cell membrane under these experimental conditions. For panel B, Dunn's post hoc test: ****, $P < 0.0001$. n = 20, 37, and 38 cells for MCF-10A, HCC-1395, and MDA-MB-231, respectively.

(red channel) was higher in the nucleolus region than in the nucleus (Fig 2A, right). Moreover, the orthogonal view of the merged confocal image clearly shows the scattered distribution of CB2R in the nucleus, whilst this protein accumulates densely within the nucleolus (Fig 2B).

Next, to more precisely investigate the accumulation of CB2R within the nucleoli, we transfected a genetically encoded fluorescent marker, the 3xnls-mTurquoise2. This construct possesses the strong nuclear location signal 3xnls variant, which enhances the labeling of nucleoli [14]. Fig 2C shows the confocal microscopy image of a 3xnls-mTurquoise2-transfected cancer cell, in which three nucleoli (gray channel) can be easily observed inside one nucleus (Fig 2C, left). In fact, the CB2R signal detected by anti-CB2 antibodies (red channel) was distributed occupying the same region of the nucleoli (Fig 2C, middle and right). The fluorescence intensity of each optical slice and channel was plotted against the position on the Z-axis and normalized to its maximum value: the well overlapping between 3xnls-mTurquoise2-nucleolar and CB2R-Alexa 647 curves indicates co-localization of the receptor in the nucleolar structure (Fig 2D).

Moreover, we performed a comparative analysis of CB2R expression between mitotic and interphase cancer cells (Fig 3). Interestingly, it was noticed that in cells undergoing mitosis, i.e., the cell phase when the nuclear and nucleolar structures are disassembled, the red labeling of CB2R was greatly reduced, appearing very dispersed throughout the cell (Fig 3A). Thus, mitotic cells exhibited a significantly lower receptor expression in comparison with interphase cells in both TNBC cell lines ($P < 0.0001$; Mann-Whitney U test) (Fig 3B), suggesting that nucleolus integrity is determinant for the sequestration of CB2R into this cell structure.

## Nucleolar sequestration of CB2R is reversed by the specific agonist HU-308

It has been shown that pharmacological activation of CB2R leads to anti-tumoral effects [16–19]. Here, we found that treating cancer cells with the highly selective CB2R agonist, HU-308, induced a redistribution of the CB2R signal away from the nucleoli (Fig 4). Indeed, the red labeling of the receptor in the nucleoli was significantly lower in the presence of this agonist

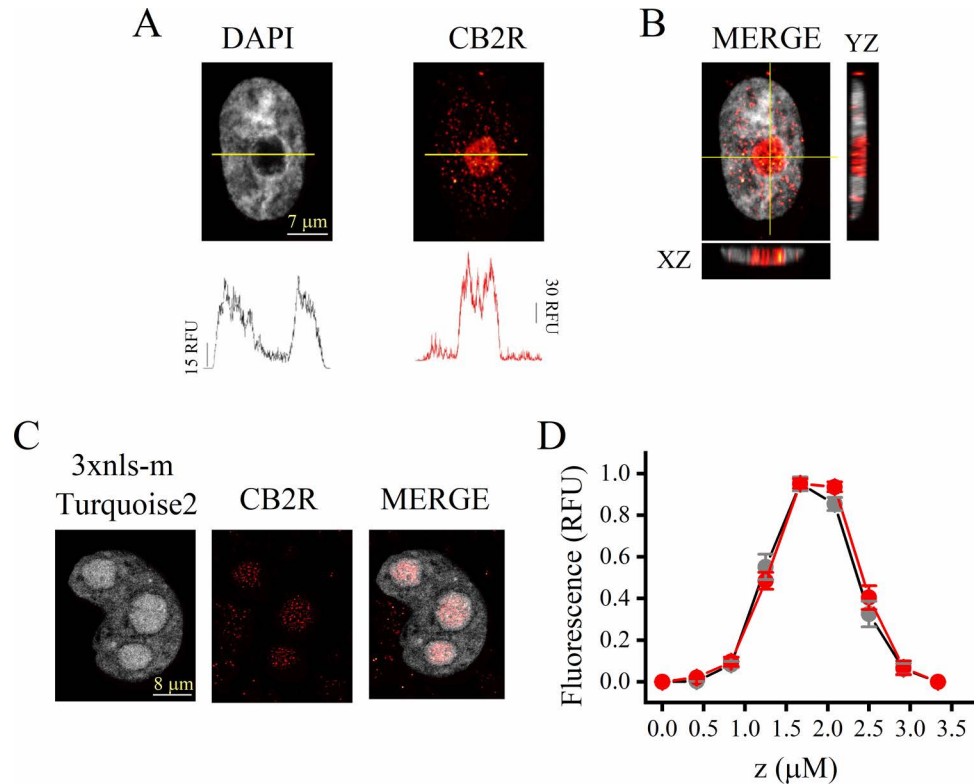

**Fig 2. Overexpression of CB2R in TNBC cell lines occurs mainly in the nucleolus. A,** Representative high magnification micrographs focused on one nucleus of double-stained cells (TNBC cell line HCC-1395) and displaying the fluorescence intensity distribution (bottom traces) from DAPI and CB2R channels at the indicated transects across both the nucleus and nucleolus (yellow lines). **B,** Merged image of the same nucleus as in panel A depicting orthogonal projections of Z-stacks (yellow lines), making evident the predominance of nucleolar localization of CB2R in TNBC cells. **C,** Illustrative images of double-stained cancer cells (line HCC-1395), labeled with the genetically encoded fluorescent marker 3xnls-mTurquoise2 and the red Alexa-Fluor 647 (for CB2R). **D,** Comparative analysis of fluorescence intensity of each channel plotted versus the Z-axis position, which was normalized to the maximal value for each signal ($n=11$).

at the two concentrations tested, 0.1 and 10 μM, compared with vehicle-treated cells for both cancer cell lines studied ($P<0.0001$; Kruskal-Wallis test) (Fig 4B). At the same time, the HU-308 treatment increased the presence of CB2R in the nucleoplasm ($P<0.0001$; Kruskal-Wallis test) (Fig 4C) and in the cytoplasm ($P<0.0001$; Kruskal-Wallis test) (Fig 4D).

### Specific CB2R agonist HU-308 decreases migration and proliferation of TNBC cells

Next, we investigated the effects of HU-308 on cell migration of TNBC cells using the wound-healing assay. After generating the cell-free area, the closure of the wound was monitored through live-cell imaging for a 36-h period, by capturing images every 3 h. Figure 5 shows that the CB2R-specific agonist HU-308 (10 μM), significantly slowed cell migration of cancerous cells ($P<0.05$–0.0001; unpaired *t*-test), being more evident in the metastatic cell line MDA-MB-231. Finally, using the EdU assay, we compared the proliferation rate of cells in the presence and absence of HU-308 (10 μM). Compared with untreated cells, HU-308 inhibited the proliferation of TNBC cells (Fig 6), although this effect was only significant in MDA-MB-231 cells ($P<0.001$; unpaired *t*-test) (Fig. 6C).

### Discussion

Several *in vitro* and *in vivo* studies have revealed that cannabinoid receptors are involved in tumor progression and that CB2R agonists have antitumor properties against various types of cancer, including the most aggressive and with

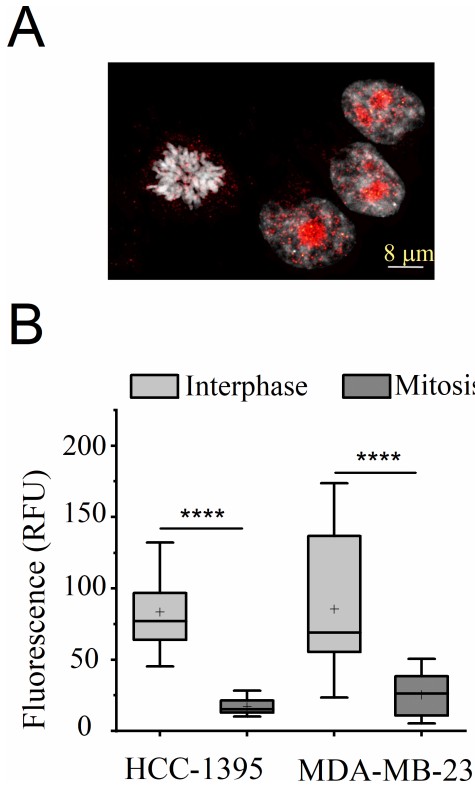

**Fig 3. CB2R sequestration is no longer observed during mitosis. A**, Confocal image displaying the distribution of the CB2R mark (red labeling) in the metastatic TNBC line cells, MDA-MB-231, during mitosis and interphase. **B**, Comparative fluorescence intensity profile of CB2R in mitotic and interphase TNBC cells. Mann-Whitney U test: ****, $P < 0.0001$. $n = 11$ and 10 for mitotic HCC-1395 and MDA-MB-231 cells, respectively; whereas $n = 37$ and 38 for interphase HCC-1395 and MDA-MB-231 cells, respectively.

the worst prognosis, the TNBC [20–22]. Likewise, cumulative evidence supports that CB2R is overexpressed in breast cancer and associated with poor prognosis [23,24]. In line with these reports, we found that CB2R is overexpressed in both early-stage I (HCC-1395) and highly metastatic (MDA-MB-231) cell lines, implying that this receptor is deregulated from the very beginnings of the pathology. Surprisingly, most of the CB2R overexpression in cancer cells was observed in the cell nucleus, specifically in the nucleolus, and treatment of cancer cells with the specific agonist HU-308 removed the CB2R from this cellular organelle.

G protein-coupled receptors (GPCRs) have traditionally been considered cell membrane signaling proteins, but cumulative evidence shows that these receptors can also be localized in intracellular compartments, including the nucleus, endosomes, endoplasmic reticulum, Golgi apparatus, and mitochondria [25]. Notably, for various intracellular GPCRs have been disclosed interesting functions and/or potential roles in several pathologies, such as cancer [26]. CB2R, a GPCR member of the endocannabinoid system, signals from both the plasma membrane and inside the cell, either by coupling to Gi/o or Gq proteins, respectively [12,13]. The subcellular localization of CB2R has been found in endolysosomes, and its physiological role is associated with Gq-eliciting $Ca^{2+}$ signaling [12]. Therefore, the nuclear presence of CB2R in TNBC cells is not unusual because many nuclear GPCRs have been reported to date [27], but what is interesting is the expression of this receptor in the nucleolus, which was confirmed using several analytical and experimental approaches. Furthermore, the CB2R signal observed in the nucleoli of interphase cells virtually disappeared when cells underwent mitosis, implicating that disassembling of the nucleolar structure during this cell cycle phase affects the

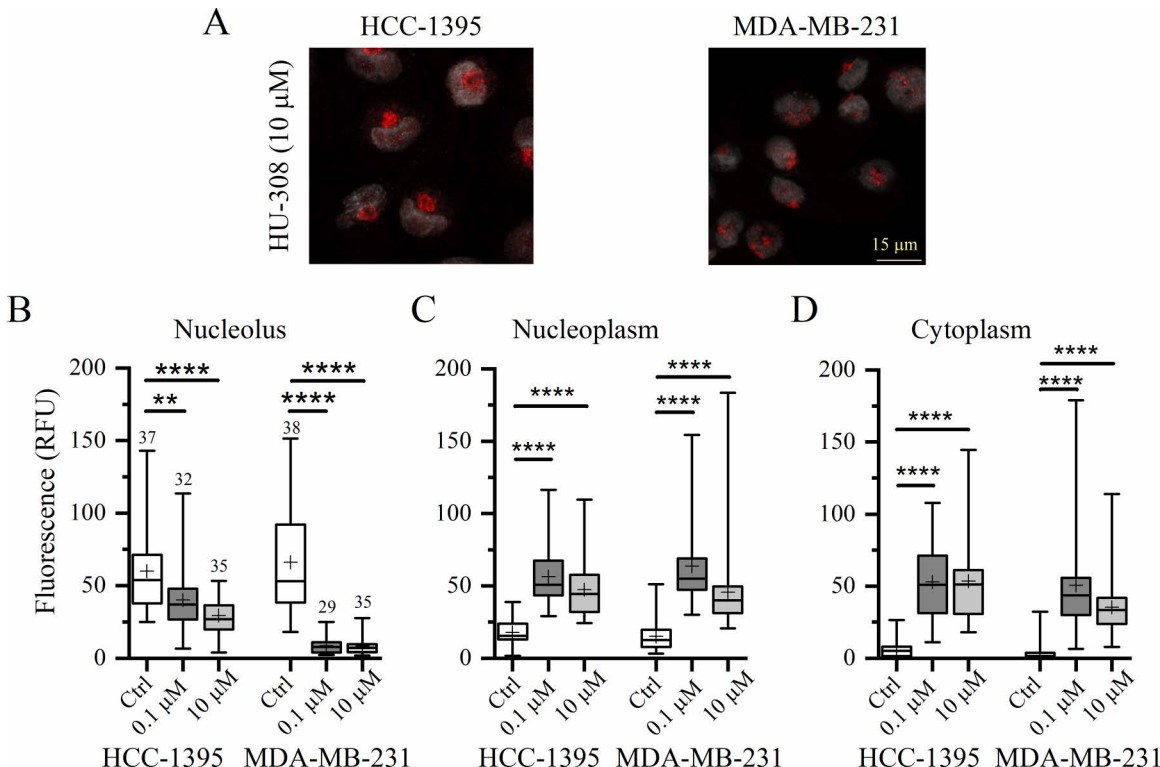

**Fig 4. Treatment of TNBC cells with HU-308 releases CB2R from the nucleoli. (A)**, confocal micrographs of cancer cell lines after 24 hours in the presence of 10 μM of the selective CB2R agonist, HU-308. Comparative analysis of red channel fluorescence intensity in the nucleolus **(B)**, nucleoplasm **(C)**, and cytoplasm **(D)**. Dunn's post hoc test: **, $P < 0.01$ and ****, $P < 0.0001$. The number of cells evaluated is indicated at the top of the whisker in the box-and-whisker plots of panel B, which are the same for panels C and D.

CB2R protein retention in this organelle, that is to say, the presence of CB2R in the nucleolus requires an intact nucleolar structure. As known, the canonical function of the nucleolus is the biogenesis of ribosomes, and the size and number of nucleoli have been used as biomarkers of tumor development [28,29]. However, during the last two decades, nucleolar proteomics studies have identified a large number of nucleolar proteins (~4,500); among these, over 30% are not involved in ribosome synthesis [30–32]. These "non-canonical nucleolar function proteins" have been associated with several roles, including control of cell cycle and proliferation, cell death, and energy production [33–36]. A non-canonical function of the nucleolus is its ability to sense diverse stresses, such as heat shock, nutrient deprivation, oncogene activation, and oxidative and genotoxic stress [37–40]. Moreover, ribosomal and other nucleolar proteins have been proposed to be involved in the cancer process by storing key tumor-associated proteins in the nucleolus [33,41,42]. This capacity to temporarily immobilize proteins within the cell nucleolus under stress conditions is termed *nucleolar sequestration* [40]. Several evidences suggest that the nucleolus also functions as a storage site for many misfolded proteins, which can be refolded when the stress is removed [34,40]. Nevertheless, the precise mechanism governing nucleolar sequestration is not yet well understood, not to mention for a GPCR, such as CB2R.

Another interesting finding in this work was that exposure of cancer cells to the specific CB2R agonist, HU-308, induced a redistribution of the receptor signal, decreasing its presence in the nucleoli while increasing it in the nucleoplasm and cytoplasm, with a more apparent effect on MDA-MB-231 cells. Furthermore, this CB2 agonist reduced the

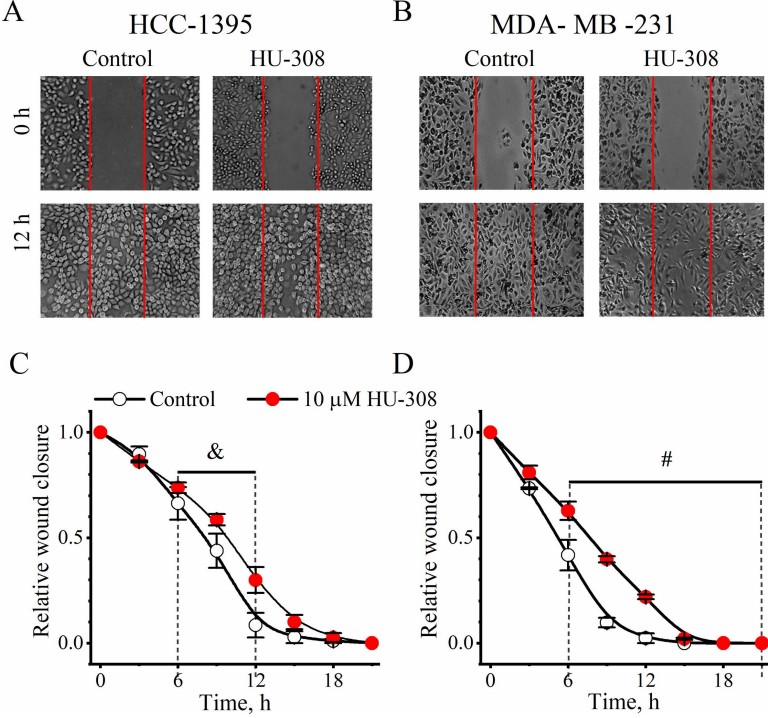

**Fig 5. Effect of the agonist HU-308 on the migration of TNBC cells, assessed with the wound healing assay.** Representative images of the scratch/wound in HCC-1395 **(A)** and MDA-MB-231 cells **(B)** at 0 and 12 h. The red lines delimit the edge of the wound at time 0 h. Quantitative analysis of wound closure over the 24 h-period (for simplicity, only the 0-21 h lapse is shown) for HCC-1395 **(C)** and MDA-MB-231 cells **(D)**. For B and C, unpaired $t$-test: &, $P < 0.05$ - 0.01; #, $P < 0.05$ - 0.0001.

biological processes of proliferation and migration of cancerous cells, also with a more evident effect on highly metastatic cells, suggesting that the greater drug-induced CB2R nucleolar removal could be linked with a more anti-cancer action of HU-308. A similar concentration of this agonist (10 µM)) was effective on the proliferation of a colon cancer cell line [43]. The specific removal of CB2R from the nucleoli of TNBC cells induced by HU-308 could also explain the antiproliferative and anticancer properties of various types of cannabinoids (phytocannabinoids, endocannabinoids, and synthetic cannabinoids) against this pathology. It should be mentioned that several polyphenolic compounds, such as curcumin and three components of poplar-type propolis, pinostrobin, pinocembrin, and pinobanksin, share a similar molecular structure with cannabinoids (including HU-308) and also have antitumor properties [44–46]. In fact, recent evidence suggests that curcumin is a CB2R agonist [47]. However, further studies are needed to determine: 1) whether all these potential anticancer agents are indeed CB2R agonists; 2) If different CB2R agonists are able to reverse CB2R nucleolar sequestration in TNBC; and 3) the overall role of this process in these tumor cells.

Although several GPCRs have been found in the nucleolar proteome [30,31], as far as we know, there is no report about the presence of a cannabinoid receptor within the nucleolus of cancer cells. Certainly, our findings are limited to TNBC cell lines, but we deem this particular feature observed from the initial to metastatic stages could be a potential early diagnostic marker for TNBC. However, it remains to be elucidated whether CB2R nucleolar sequestration is a specific quality of TNBC cell lines and, most importantly, whether it occurs in human TNBC tissue, which would allow a better prognosis by initiating early treatment of this pathology. Additionally, this intriguing finding raises several additional questions that should be addressed in future research, including identifying the function of CB2R in the nucleolus and whether these receptors are capable of signaling.

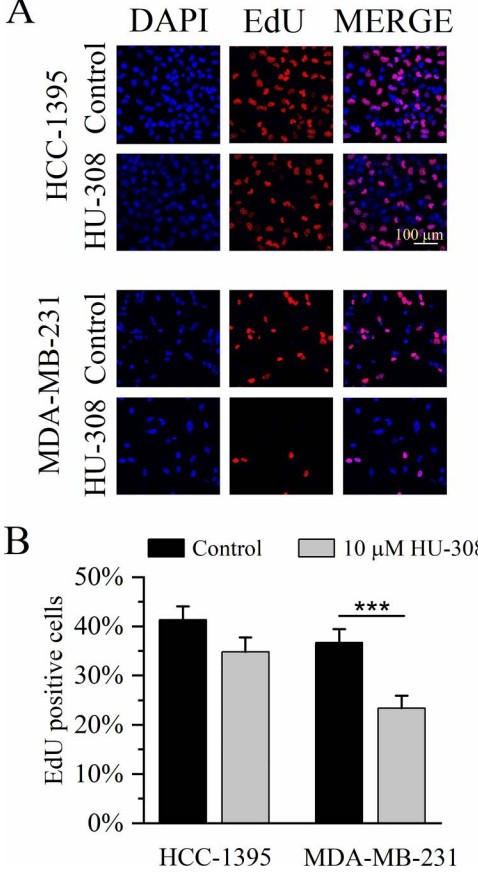

**Fig 6. The CB2 receptor agonist HU-308 reduces the proliferation of TNBC metastatic cells. A**, Representative images of TNBC cell lines stained with EdU (purple, denoting proliferating cells), DAPI (blue), and the merge of both channels (violet). B, Bar graphs of the cell proliferation, obtained as a percentage from the relative fold of EdU-positive cells. Experiments were conducted in triplicate. For B, unpaired *t*-test: ***, *P* < 0.0001.

In conclusion, this is the first study unveiling not only the presence but a vast accumulation of a CB2R within the nucleolus of TNBC cells. This interesting feature is reversed by the specific CB2 agonist HU-308, whose effect is associated with a reduction in both cell proliferation and migration.

## Supporting information

**S1 Fig. The blocking peptide completely abolished the staining of Anti-CB2R antibody.**
(TIF)

**S2. Raw data.**
(XLSX)

## Acknowledgments

We are grateful to Miguel Angel Flores Virgen for technical assistance.

## Author contributions

**Conceptualization:** Ricardo Antonio Navarro-Polanco.

**Data curation:** Linley P. Prado-Celis, Rodrigo Zamora-Cardenas, Enrique A. Sanchez-Pastor.

**Formal analysis:** Linley P. Prado-Celis, Rodrigo Zamora-Cardenas, Javier Alamilla, Tania Ferrer, Eloy G. Moreno-Galindo.

**Funding acquisition:** Javier Alamilla, Eloy G. Moreno-Galindo, Ricardo Antonio Navarro-Polanco.

**Investigation:** Eloy G. Moreno-Galindo, Ricardo Antonio Navarro-Polanco.

**Writing – original draft:** Eloy G. Moreno-Galindo, Ricardo Antonio Navarro-Polanco.

**Writing – review & editing:** Linley P. Prado-Celis, Rodrigo Zamora-Cardenas, Javier Alamilla, Enrique A. Sanchez-Pastor, Tania Ferrer, Eloy G. Moreno-Galindo, Ricardo Antonio Navarro-Polanco.

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
