## [Decision Letter · Decision Letter 0]

26 Dec 2024

PONE-D-24-47936Nlucleolar Sequestration of Cannabinoid Type-2 Receptor in Triple-Negative Breast Cancer CellsPLOS ONE

Dear Dr. Navarro-Polanco,

Thank you for submitting your manuscript to PLOS ONE. After careful consideration, we feel that it has merit but does not fully meet PLOS ONE’s publication criteria as it currently stands. Therefore, we invite you to submit a revised version of the manuscript that addresses the points raised during the review process.

We look forward to receiving your revised manuscript.

Kind regards,

Muhammad Ahmad

Academic Editor

PLOS ONE

Journal Requirements:

3. Thank you for stating the following financial disclosure: This work was funded by Consejo Nacional de Humanidades, Ciencia y Tecnologia, Mexico: to R.A.N-P., E.G.M-G., and J.A. (Grant No. FC2015-1-121) and Infrastructure Grant No. 321696. L.P.P-C. and R.Z-C. were supported by Doctoral Fellowships (No. CVU 933434 and 618681, respectively) from CONAHCYT, Mexico.

Reviewers' comments:

Reviewer's Responses to Questions

**Comments to the Author**

1. Is the manuscript technically sound, and do the data support the conclusions?

Reviewer #1: Yes

Reviewer #2: Yes

Reviewer #3: Yes

Reviewer #4: Yes

2. Has the statistical analysis been performed appropriately and rigorously? 

Reviewer #1: Yes

Reviewer #2: Yes

Reviewer #3: Yes

Reviewer #4: N/A

3. Have the authors made all data underlying the findings in their manuscript fully available?

Reviewer #1: Yes

Reviewer #2: Yes

Reviewer #3: No

Reviewer #4: No

4. Is the manuscript presented in an intelligible fashion and written in standard English?

Reviewer #1: Yes

Reviewer #2: Yes

Reviewer #3: Yes

Reviewer #4: Yes

5. Review Comments to the Author

Reviewer #1: Thank you so much for your responses to the Reviewers' comments. I have no further concerns at the moment. Congrats on your work. I believe that this article is now acceptable for publication in Plos One.

Reviewer #2: The current version of the paper has been already reviewed by three other reviewers in the first round of the review process and the authors addressed all the comments of the previous round properly and the paper can be accepted for publication.

Reviewer #3: This study investigates the expression and subcellular localization of the cannabinoid type-2 receptor (CB2R) in triple-negative breast cancer (TNBC) cell lines at different stages of disease progression and compares them to a non-tumoral mammary epithelial cell line. The authors demonstrate that CB2R is significantly overexpressed in TNBC cells and predominantly accumulates in the nucleoli. Interestingly, activating CB2R with a specific agonist (HU-308) redistributes the receptor away from the nucleoli to the nucleoplasm and cytoplasm, and this redistribution is accompanied by reduced cell migration and proliferation. The authors propose that nucleolar sequestration of CB2R could serve as a diagnostic marker and potentially influence TNBC progression.

The work is technically sound, well-reported, and potentially impactful for TNBC research.

The authors should address the points about data accessibility, provide a bit more mechanistic speculation or references, and refine the discussion on the translational implications. After these minor improvements, the manuscript would be suitable for publication in PLOS ONE.

A number of minor revisions regarding discussion part are suggested before to proceed publishing:

1. While the study reveals nucleolar accumulation of CB2R, the mechanism underlying this sequestration remains unclear. Elaborating on potential nucleolar retention signals or interacting proteins would strengthen the manuscript’s impact. This may not be essential for publication, but providing a brief discussion of possible mechanisms or future directions would be helpful.

2. Although HU-308 reduces nucleolar CB2R and decreases proliferation and migration, it appears more effective in metastatic MDA-MB-231 cells than in early-stage HCC-1395 cells. A brief exploration or discussion as to why this differential response occurs would be valuable.

3. Additional discussion on the potential clinical implications or how one might translate these findings into diagnostic or therapeutic approaches could strengthen the relevance.

4. The manuscript would benefit from a clear statement on data availability, including any raw images or numerical data. This will support reproducibility and compliance with PLOS ONE’s policies

5. In Figure legends, more details on sample sizes and exact statistical tests used for each comparison would help clarify the results.

Reviewer #4: What is conclusion of that study??

Elaborate the statistical model used?

There is need to define the study design briefly?

Image J is used where it is impleneted in fig ….?

Go through these articles to support the manuscript and further understanding

https://www.pvj.com.pk/pdf-files/23-430.pdf

https://www.pvj.com.pk/pdf-files/24-471.pdf

https://pvj.com.pk/pdf-files/23-052.pdf

https://www.ijvets.com/pdf-files/24-427.pdf

6. PLOS authors have the option to publish the peer review history of their article (what does this mean? ). If published, this will include your full peer review and any attached files.

**Do you want your identity to be public for this peer review?** For information about this choice, including consent withdrawal, please see our Privacy Policy .

Reviewer #1: **Yes: ** Alex Mabou Tagne

Reviewer #2: No

Reviewer #3: No

Reviewer #4: No

---

## [Author Response · Author response to Decision Letter 1]

10 Feb 2025

RESPONSES TO THE REVIEWERS’ COMMENTS

We thank the reviewers, both the previous and the new ones, for their comments on our study and the suggestions to improve our MS. As in the preceding resubmission, changes in the MS are shown in red font, while responses (R) to your specific comments are denoted in blue font.

Reviewer #1: Thank you so much for your responses to the Reviewers' comments. I have no further concerns at the moment. Congrats on your work. I believe that this article is now acceptable for publication in Plos One.

Reviewer #2: The current version of the paper has been already reviewed by three other reviewers in the first round of the review process and the authors addressed all the comments of the previous round properly and the paper can be accepted for publication.

Reviewer #3: This study investigates the expression and subcellular localization of the cannabinoid type-2 receptor (CB2R) in triple-negative breast cancer (TNBC) cell lines at different stages of disease progression and compares them to a non-tumoral mammary epithelial cell line. The authors demonstrate that CB2R is significantly overexpressed in TNBC cells and predominantly accumulates in the nucleoli. Interestingly, activating CB2R with a specific agonist (HU-308) redistributes the receptor away from the nucleoli to the nucleoplasm and cytoplasm, and this redistribution is accompanied by reduced cell migration and proliferation. The authors propose that nucleolar sequestration of CB2R could serve as a diagnostic marker and potentially influence TNBC progression.

The work is technically sound, well-reported, and potentially impactful for TNBC research.

The authors should address the points about data accessibility, provide a bit more mechanistic speculation or references, and refine the discussion on the translational implications. After these minor improvements, the manuscript would be suitable for publication in PLOS ONE.

A number of minor revisions regarding discussion part are suggested before to proceed publishing:

1. While the study reveals nucleolar accumulation of CB2R, the mechanism underlying this sequestration remains unclear. Elaborating on potential nucleolar retention signals or interacting proteins would strengthen the manuscript’s impact. This may not be essential for publication, but providing a brief discussion of possible mechanisms or future directions would be helpful.

R = Yes, you are right that including the mechanism by which CB2R is sequestered in the nucleolus of cancerous cells would increase the manuscript’s impact, although the precise mechanism for the nucleolar sequestration is unknown, especially for a GPCR, being this work the first report on this topic. Therefore, we don´t feel comfortable speculating (even a bit) about a virtually unknown mechanism. In the Discussion, we added two lines mentioning this limitation (lines 6-8, page 17). We hope that this point of view is not problematic for you.

2. Although HU-308 reduces nucleolar CB2R and decreases proliferation and migration, it appears more effective in metastatic MDA-MB-231 cells than in early-stage HCC-1395 cells. A brief exploration or discussion as to why this differential response occurs would be valuable.

R = In the Discussion section, we delineated a plausible reasoning for the more pronounced impact of HU-308 on MDA-MB-231 cells, focusing on the removal of CB2R from the nucleolus (lines 12, 14-17, page 17).

3. Additional discussion on the potential clinical implications or how one might translate these findings into diagnostic or therapeutic approaches could strengthen the relevance.

R = Done (lines 13-15, page 18).

4. The manuscript would benefit from a clear statement on data availability, including any raw images or numerical data. This will support reproducibility and compliance with PLOS ONE’s policies

R = We included the data availability statement in the document along with an Excel file in the Supporting Information section with all the raw data (if I'm not mistaken, this file can be observed after the figures in the Metadata Journal file).

5. In Figure legends, more details on sample sizes and exact statistical tests used for each comparison would help clarify the results.

R = The statistical test and the sample size are indicated in the Figure legends.

Reviewer #4:

What is conclusion of that study??

R = Thanks for your observation. We added a conclusion at the end of the Discussion. To do this, we had to modify a little bit the first lines of the previous paragraph (lines 7-8, page 18).

Elaborate the statistical model used?

R = In the Statistical Analysis section, we are mentioning the adequate central and dispersion values according to the distribution of data, as verified with two normality tests (Shapiro–Wilk and Kolmogorov-Smirnov tests). We then applied parametric and non-parametric statistical tests/models, as required. Moreover, the sample size and the specific statistical test used are indicated in the Figure legends.

There is need to define the study design briefly?

R = Part of the study is descriptive and the part of the agonist’s effect is experimental. However, due to the basic nature/profile of the study, we judge that is not really necessary to mention this issue in the MS. We hope that there is no problem about it.

Image J is used where it is impleneted in fig ….?

R = ImageJ was used for the analysis of all images, as commented in Methods.

Go through these articles to support the manuscript and further understanding

https://www.pvj.com.pk/pdf-files/23-430.pdf

https://www.pvj.com.pk/pdf-files/24-471.pdf

https://pvj.com.pk/pdf-files/23-052.pdf

https://www.ijvets.com/pdf-files/24-427.pdf

R = Dear Reviewer, with all due respect, we tried hard to include the findings of the papers you suggest. Despite these references deal with potential anti-tumoral agents for breast cancers, unfortunately, we could not find a direct relationship with the core topics of our work: TNBC, cannabinoid receptors and nucleolar sequestration.

---

## [Decision Letter · Decision Letter 1]

12 Mar 2025

PONE-D-24-47936R1Nlucleolar Sequestration of Cannabinoid Type-2 Receptor in Triple-Negative Breast Cancer CellsPLOS ONE

Dear Dr. Navarro-Polanco,

Thank you for submitting your manuscript to PLOS ONE. After careful consideration, we feel that it has merit but does not fully meet PLOS ONE’s publication criteria as it currently stands. Therefore, we invite you to submit a revised version of the manuscript that addresses the points raised during the review process.

We look forward to receiving your revised manuscript.

Kind regards,

Muhammad Ahmad

Academic Editor

PLOS ONE

Journal Requirements:

Reviewers' comments:

Reviewer's Responses to Questions

**Comments to the Author**

1. If the authors have adequately addressed your comments raised in a previous round of review and you feel that this manuscript is now acceptable for publication, you may indicate that here to bypass the “Comments to the Author” section, enter your conflict of interest statement in the “Confidential to Editor” section, and submit your "Accept" recommendation.

Reviewer #2: (No Response)

Reviewer #3: All comments have been addressed

2. Is the manuscript technically sound, and do the data support the conclusions?

Reviewer #2: (No Response)

Reviewer #3: Yes

3. Has the statistical analysis been performed appropriately and rigorously? 

Reviewer #2: (No Response)

Reviewer #3: Yes

4. Have the authors made all data underlying the findings in their manuscript fully available?

Reviewer #2: (No Response)

Reviewer #3: Yes

5. Is the manuscript presented in an intelligible fashion and written in standard English?

Reviewer #2: (No Response)

Reviewer #3: Yes

6. Review Comments to the Author

Reviewer #2: The authors of the paper have fully addressed all the concerns of the previous round, I do not have any further comments.

Reviewer #3: Thank you for your revisions. I appreciate the efforts made to address the comments.

1-I understand your reasoning regarding the mechanism of CB2R nucleolar sequestration and agree that acknowledging this limitation in the discussion is a reasonable approach.

2-The additional discussion on the differential response of the cells provide valuable context.

3-The expanded discussion on the potential clinical implications strengthens the relevance of the findings.

4-Data has already been provided in the supplementary data. The statement is also helpful.

With these revisions, I find the manuscript suitable for publication. I have no further concerns.

Best regards,

7. PLOS authors have the option to publish the peer review history of their article (what does this mean? ). If published, this will include your full peer review and any attached files.

**Do you want your identity to be public for this peer review?** For information about this choice, including consent withdrawal, please see our Privacy Policy .

Reviewer #2: No

Reviewer #3: No

---

## [Author Response · Author response to Decision Letter 2]

27 Mar 2025

We thank to the editor for his comments on our study and the suggestions to improve our MS. As in the preceding resubmission, changes in the MS are shown in red font, while responses (R) to your specific comments are denoted in blue font.

1) The keywords should not be included in the title. Please revise accordingly.

R. Done

2) The beginning of the second paragraph of introduction is not appropriate. Consider rewording for better clarity and coherence.

R. Done

3) Improve the overall English language and revise for better readability.

R. Done

4) Check for any grammatical mistakes and correct them.

R. We have already revised and corrected the grammar of the manuscript in more detail.

---

## [Editor Report · Decision Letter 2]

10 Apr 2025

Nucleolar sequestration of cannabinoid type-2 receptor in triple-negative breast cancer cells

PONE-D-24-47936R2

Dear Dr. Navarro-Polanco,

We’re pleased to inform you that your manuscript has been judged scientifically suitable for publication and will be formally accepted for publication once it meets all outstanding technical requirements.

Kind regards,

Muhammad Ahmad

Academic Editor

PLOS ONE
---

## [Editor Report · Acceptance letter]

PONE-D-24-47936R2

PLOS ONE

Dear Dr. Navarro-Polanco,

I'm pleased to inform you that your manuscript has been deemed suitable for publication in PLOS ONE. Congratulations! Your manuscript is now being handed over to our production team.

Kind regards,

on behalf of

Mr. Muhammad Ahmad

Academic Editor

PLOS ONE